# Endoscopy in IBD: When and How?

**DOI:** 10.3390/diagnostics13223423

**Published:** 2023-11-10

**Authors:** Marco Daperno

**Affiliations:** Gastroenterology Unit, Mauriziano Hospital, 10128 Torino, Italy; mdaperno@gmail.com

**Keywords:** endoscopy, diagnosis, prognosis, central review, agreement, endoscopic scores

## Abstract

Endoscopy is an essential tool supporting inflammatory bowel disease diagnosis, and ileocolonoscopy is essential to the diagnostic process because it allows for histological sampling. A decent description of endoscopic lesions may lead to a correct final diagnosis up to 89% of the time. Moreover, endoscopy is key to evaluating endoscopic severity, which in both Crohn’s disease and ulcerative colitis is associated with worse disease outcomes (e.g., more frequent advanced therapy requirements or more frequent hospitalizations and surgeries). Endoscopic severity should be reported according to validated endoscopic scores, such as the Mayo endoscopic subscore (MES) or the ulcerative colitis endoscopic index of severity (UCEIS) for ulcerative colitis, the Rutgeerts score for postoperative Crohn’s recurrence, and the Crohn’s disease endoscopic index of severity (CDEIS) or the simplified endoscopic score for Crohn’s disease (SES-CD) for luminal Crohn’s disease activity. The measuring of endoscopic activity has become a regulatory agency requirement to increase the objective evaluation of disease activity and drug response. In recent years, the central reviewing of endoscopic videos has become a standard for clinical trials. However, the adjudication paradigm and the type of endoscopic reading may substantially affect trial outcomes, and the reproducibility of all endoscopic scores is not perfect as they require the interpretation of intrinsically subjective images. This paper reviews and discusses the available evidence on inflammatory bowel disease endoscopy.

## 1. Introduction

Endoscopy is pivotal for the diagnosis and therapeutic monitoring of inflammatory bowel disease (IBD) in both clinical practice and clinical trials.

The evolution of endoscopic machinery offers clinicians increased scope with progressively more precise and high-definition imaging, enabling them to assess the bowel in more depth using enteroscopes and wireless capsule endoscopy and providing several new endoscopic tools to achieve higher diagnostic and therapeutic goals (e.g., endomicroscopy probes or tools such as dilation balloons, self-expandable metal stents, and more).

The correct planning of endoscopy is key to using the technique at its best for the diagnosis, monitoring, and evaluation of the prognosis of IBD patients. Moreover, a new field for the use of endoscopy in IBD is the integration of endoscopic results into IBD clinical trials.

## 2. Diagnosis

The main diagnostic tool (of which the results should be integrated with the patient’s history, clinical feature interpretation, and laboratory and pathological findings) is ileocolonoscopy, which enables a reliable diagnosis in slightly less than 9 out of 10 of cases [1], allows tissue sampling to support pathological reports, and is considered the mainstay of IBD diagnosis per the current ECCO guidelines [2,3].

Crohn’s disease (CD) and ulcerative colitis (UC) may involve variable extents of the small bowel and the colon; however, very commonly, the most representative pathological lesions lie in the colon or the terminal ileum, allowing for adequate visualization, characterization, and classification during an ileocolonoscopy. In a pivotal study, Pera et al. [1] reported that, when analyzing 606 colonoscopies carried out for known or suspected IBD, the accuracy of the colonoscopies was 89%, with 7% being indeterminate diagnoses and only 4% errors, which were numerically more likely in patients displaying the most severe inflammatory activity (in that subgroup, the occurrence of errors was more than doubled, reaching 9%). The study reported that the most useful endoscopic features for a differential diagnosis between CD and UC were discontinuous involvement, anal lesions, and cobblestoning of the mucosa (increasing the likelihood of a final diagnosis of CD) compared with erosions and mucosal granularity (increasing the likelihood of a final diagnosis of UC).

According to the current ECCO guidelines, IBD diagnosis requires the integration of several clinical, imaging, and laboratory findings [2,3]. While all of these techniques are important, endoscopy, with its ability to collect biopsy specimens to support the diagnostic process, remains crucial. Moreover, the first (diagnostic) endoscopic examination contributes to a clear classification of the extent of the colon involved in the disease for ulcerative colitis, as well as the location of Crohn’s disease according to the Montreal classification [4]. It may also serve as a starting point for planning subsequent follow-ups and a priori risks of aggressive disease. If the endoscopic observation and its description are very precise, the endoscopic report can be of great value in reaching the correct final IBD diagnosis and characterization [5].

In the diagnostic process, endoscopy of the small bowel, either using capsule endoscopy (CE) or device-assisted enteroscopy (DAE), may also be needed. The evaluation of the small bowel is a strongly recommended step in Crohn’s disease, as well as when the ileocolonoscopy looks diagnostic for Crohn’s disease or when a diagnosis of IBD is unclassified [6]. In most cases, cross-sectional imaging is used to support the presence/absence of upper gastrointestinal tract involvement and extramural complications, such as abscesses, fistulas, or mesenteric hypertrophy. However, in selected cases, CE and DAE may be needed to exclude lesions relevant to the outcome, as a disease location in the jejunum or the proximal ileum is considered an adverse prognostic factor for a worse disease outcome of CD (upper gastrointestinal tract localization) [7]. Moreover, DAE can help with tissue sampling when a differential diagnosis in the small bowel includes small bowel lymphoma, adenocarcinoma, or small bowel tuberculosis, which is impossible using cross-sectional imaging and CE. Using a more diffuse diagnostic approach with CE to initially stage CD at diagnosis, or when CD is suspected, results in a higher reported risk of capsule retention (up to approximately 1 in 8 patients with known CD and 1 in 60 patients with suspected CD) [8]. Due to this increased risk, systematically performing a patency capsule test before a CE examination in known or suspected CD is strongly recommended.

In summarizing the evidence on using endoscopy in IBD diagnosis according to the questions of ‘when’ and ‘how’ in this paper’s title, endoscopy is essential for IBD diagnosis.

**When?** When a subject is suspected of being affected by IBD based on his/her clinical features, endoscopy is an essential diagnostic tool due to its very high intrinsic accuracy and specificity. Because of the disease’s high pretest probability, only a very accurate diagnostic technique is entitled to exclude or confirm a final diagnosis of IBD.

**How?** Ileocolonoscopy with segmental biopsies is the preferred technique. The careful reporting of lesions, with the location and severity from the first examination, may impact the subsequent diagnosis and management of IBD patients [5]. Capsule endoscopy and disease-assisted enteroscopy may be chosen in selected cases [6].

## 3. Severity Assessment

IBD endoscopic severity affects disease outcomes. This statement is supported by a large body of evidence, wherein the most severe endoscopic lesions are associated with the worst disease outcomes (an increased likelihood of surgery or no response to medical treatments) [9,10,11], while amelioration or healing of severe endoscopic lesions is associated with the best disease outcomes (less common clinical relapses, the least need for advanced treatments, and/or the lowest surgical risks) [12,13,14,15,16,17,18].

Crohn’s disease patients with deep ulcerations involving a large part of a colonic segment had a 5–6-fold higher risk of needing a colectomy (and, therefore, sometimes also needing an ostomy) compared with patients without such lesions who were equally clinically active [9]. In that study, the occurrence of severe endoscopic lesions resulted in being a risk factor for surgery at least as important as experiencing severe clinical activity (defined by a Crohn’s disease activity index (CDAI) of greater than 288 points) or in being undertreated (defined by the absence of immunosuppressive treatment in the follow-up) [9].

Severe ulcerative colitis patients with severe endoscopic lesions were at a significantly higher risk of non-response to steroid treatment, leading to a 40-fold increase in the risk of urgent colectomy in a no-rescue treatment era [10]. Carbonnel and colleagues identified four types of severe lesions associated with a high risk of non-response to medical treatments, irrespective of exhibiting similar clinical activity:Deep and extensive ulcerations, bounded by swollen mucosa;Mucosal detachment, which can be demonstrated by inserting the biopsy forceps under the edge of the ulceration;Well-like ulcerations, visible as very deep ulcerations with a small diameter;Large mucosal abrasions, formed by the junction of several deep ulcerations.

In addition to the paper stating the efficacy of steroids in lowering the mortality of severe ulcerative colitis [18], there was an endoscopic observation of the effective drug’s healing effects.

When Crohn’s disease patients, usually in clinical remission (generally speaking, asymptomatic), are evaluated via early colonoscopy as early as 1 year after surgery, the severity of endoscopic lesions is highly associated with different risks of long-term clinical (and surgical) recurrence [11]. The Rutgeerts criteria for post-surgical recurrence, well replicated in several later studies, allow for the subgrouping of patients with remarkably different disease recurrence risks at 5 years [11]:Patients with no recurrent lesions in the neoterminal ileum (i0; see Figure 1A) and those with up to five aphthous lesions in the neoterminal ileum (i1) had a clinical recurrence risk of 10% at 8 years;Patients with more than five aphthous lesions in the neoterminal ileum with a normal mucosa between them, or with ulcers isolated from the anastomotic line (<1 cm), classified as i2, had a clinical recurrence risk of 40% at 5 years;Patients with diffuse aphthous ileitis, with diffusely inflamed mucosa in between aphthae (classified as i3; see Figure 1B), had a clinical recurrence risk of 80% at 6 years;Finally, patients with diffuse inflammation with already larger ulcers, nodules, and/or lumen narrowing (classified as i4; see Figure 1C) had a clinical recurrence risk of 100% at 4 years.

Therefore, early endoscopic lesions (detected 3–12 months after surgery) in asymptomatic patients allow for a reliable categorization of the subsequent individual risk of clinical relapse if the disease is not treated. Only i0 and i1 patients are likely to not relapse. However, later observation of the same endoscopic features is not prognostically relevant more than 3 years after the index surgery [19]. Therefore, one early examination is advisable to stratify the individual recurrence risk, while subsequent endoscopic examinations carry little or no prognostic value and should be performed only for clinical practice, cancer surveillance, or study purposes.

Moreover, the prognostic relevance of the Rutgeerts score was developed and confirmed only for ‘curative’ ileocolonic resections, and the score should not be used for other resections (e.g., purely ileal, jejunal, or colonic resections) or if strictureplasties are carried out instead of resections.

Another issue regarding the current use and adaptation of the original Rutgeerts score [11] concerns the different shapes of ileocolonic anastomoses; the original score was developed and validated based on end-to-end enteral anastomoses, which were commonly carried out in the 1980s and 1990s. Later on, the surgical techniques and machinery evolved, and to reduce the risks of recurrence, surgeons developed new shapes with larger anastomotic lines, usually based on the side-to-side technique and its variations. The current understanding is that, regardless of the shape of the anastomosis, there is a hierarchy of lesions that recur after curative ileocolonic resection; lesions occurring in the neoterminal ileum, at the ileal inlet, or in the ileal body are more likely to be prognostically relevant compared with lesions limited to the anastomotic line (which are more likely to be ischemic due to the suturing process and, therefore, less evolutive) or to the ileal or colonic blind loop (which are more likely to be due to bacterial overgrowth and are unlikely to affect anastomotic patency) [20]. Meanwhile, the new anastomotic shaper requires special attention from the endoscopist and the clinician to correctly assign lesions to their correct anatomical locations and determine their real prognostic meanings (Figure 2) [20].

Besides the evidence that patients displaying more severe endoscopic features will have a worse prognosis, several studies in different clinical settings have shown that different degrees of healing of endoscopic lesions are associated with different clinical outcomes, irrespective of clinical remission. Patients with ulcerative colitis treated with a first steroid course at their first disease flare may have worse outcomes in terms of long-term surgical or immunosuppressive risks if there is still some degree of endoscopic activity compared with patients with full endoscopic and clinical remission [12]. The same was shown for Crohn’s disease patients in remission. If endoscopic activity remains, the risks of developing new fistulas are higher, as is the likelihood of requiring more infliximab [13]. In many studies on Crohn’s disease and ulcerative colitis, healing or failing to heal endoscopic lesions identify distinct IBD patient subgroups with different prognoses: healers have a more favorable disease course even if major treatments are stopped or in the absence of more advanced treatments [15,16,17]. This may reflect the ability of endoscopic features to identify patients with different prognoses or the ability of drugs to induce variations in endoscopic features associated with different outcomes. This point is difficult to dissect via clinical trials.

There is a need to explore and document the severity of endoscopic activity and the degrees of healing of lesions. As such, using scoring systems for endoscopic activity in IBD is of paramount importance in both clinical practice and trials to most reliably quantify endoscopic activity and its variations.

Even if clinicians and endoscopists have limited knowledge about the most common scoring systems, in clinical practice, their routine use may contribute to reporting endoscopic features more reproducibly [21], and their adoption may be spread via common educational programs [21,22].

Considering ulcerative colitis, the lesions visible in a rectosigmoidoscopy and a full colonoscopy are the same [23]. Proximal lesions not visible in a distal endoscopy substantially change the results highlighted in a distal endoscopy in less than 4% of cases. Since no bowel cleansing is needed for a distal endoscopy, the examination is faster and there is less discomfort for the patient. Limiting an ulcerative colitis endoscopic evaluation to only rectosigmoidoscopy is a reasonable option if the goal is to assess endoscopic activity.

In clinical trials, it is even more essential to report endoscopic activity using scoring systems to outline relevant differences after a therapeutic intervention in the best possible fashion [24]. The most commonly used and/or more validated different endoscopic scores are detailed as follows:The Mayo endoscopic subscore (MES) [25] and the ulcerative colitis endoscopic index of severity (UCEIS) [26] for ulcerative colitis;The Rutgeerts score [11] for post-surgical Crohn’s disease recurrence;The Crohn’s disease index of severity (CDEIS) [27] and the simple endoscopic score for Crohn’s disease (SES-CD [28], with the subsequent modified multiplier MM-SES-CD [29]) for overall Crohn’s disease.

The MES [25] is easy and straightforward and is a regulatory agency requirement. Nonetheless, it has limitations. For example, it refers to the worst lesions; therefore, it is unprecise in the most severe patients (having one superficial 1 cm ulcer leads to the same MES 3 class of having a completely ulcerated colon in every segment with deep and large ulcers). Moreover, it lacks accuracy concerning the distribution of lesions and the evaluation of partial ameliorations, which may be relevant to appreciate new treatment effects. In patients with some degree of healing, the MES does not mention the relevant ameliorations. A theoretical patient who transitioned from having a colon full of ulcers to only a few ulcers in the rectum will be classified as MES 3 at both the initial and final stages, even if remarkable amelioration occurred. Within MES 0, there are probably subgroups with different microscopic inflammation, leading to different relapse and cancer risks in the long term. Attempts to better clarify deep remission at the endoscopic level have been proposed to overcome this limitation [30].

The MES 0 degree includes both completely normal-appearing mucosa and areas of healing (with scarring (see Figure 3A) or with post-inflammatory polyps, regardless of erosions on their tops), but no erythema, friability, erosions, or ulcers are admitted, and the vascular pattern should be clearly recognized. The MES 1 degree is usually considered in continuity with MES 0. However, a higher amount of mucosal edema results in the blurring and partial obliteration of the vascular pattern, and the mucosa looks coarser, with more fragmented light scattering, sometimes with mild erythema distributed evenly across the area (Figure 3B). MES 1 is perceived as a transition from MES 0 to MES 2, but it is remarkably less inflamed than the latter. The main landmarks to increase endoscopic activity to the MES 2 degree are diffuse erythema with friability (which is not easily assigned in still images or videos recorded elsewhere) and erosions (Figure 3C). The features leading to the highest degree, MES 3, are either ulcers or spontaneous bleeding (they should not coexist to be scored as MES 3). Ulcers usually occur in areas of erythema, friability, and spontaneous bleeding, which have to be observed live to be recognized and are the consequence of extreme friability and may be present before the scope intubation of a segment. Typical UC endoscopic lesions are uniform, but after treatment or in long-standing colitis, these areas may coexist with more severe lesions (MES 2–3) and areas with complete or near-complete remission (MES 0–1). In latter cases, however, the endoscopist should score the MES according to the most severe endoscopic lesions observed, even if present in only a few cm of the colon out of several cm explored overall.

The UCEIS [26] is significantly more precise. It is formally prospectively validated and allows for a clearer characterization of the most important lesions. However, it has the same limitations as the MES when the extent of lesions and their density must be considered. Some attempts have been made to overcome the score’s intrinsic issues.

The Rutgeerts score [11] is easily assigned and profoundly impacts clinical outcomes at the individual level. This score is commonly used by endoscopists. However, it also has limitations. For instance, it refers exclusively to ileocolonic resections with end-to-end anastomosis, not being significant or validated in the case of different anastomoses. Moreover, some lesions seem to carry different meanings and prognostic values. It was reported that isolated anastomotic lesions have less impact on disease outcomes than afferent limb lesions. Finally, there is also a hierarchical approach to lesion scoring using the Rutgeerts score but features sometimes overlap between different degrees. Examples of using the Rutgeerts score are reported in Figure 1.

The CDEIS [27] and the SES-CD [28] are much more analytical and complex than the MES or Rutgeerts score. They have been validated in live examinations and have prognostic implications [9,13,29]. Their adoption in clinical practice is still limited by the ability of endoscopists to systematically record scores, but agreement on their scoring is very high [21,31], including between inexperienced endoscopists. Moreover, they have become a regulatory agency requisite since endoscopic healing was listed as an objective therapeutic endpoint in clinical trials.

The CDEIS [27] and SES-CD [28] require the endoscopist to explore and report on the five ileocolonic segments: the ileum, right colon, transverse colon, left colon, and rectum. The absence of one or more segments (due to surgical resection or the impossibility to proximally reach it due to a stenosis that cannot be passed) gives a null score for that/those segments, impacting the algorithm of the CDEIS, while the SES-CD is not considered for severe cases. The minimal number of explorable and considerable segments in the original papers was two [27,28] in the case of ileorectal anastomosis resulting after a total colectomy. Even if an examination via stomas is not considered, using the CDEIS and SES-CD is easily expandable to such clinical conditions (either referring to a segmental score for the ileal area upstream of an ileostomy or to a complete score for an endoscopy above a colostomy).

Lesions considered using both the mentioned scores include ulcers, the percentage of a segment covered by ulcers, narrowing/stenosis, all inflammatory features of Crohn’s disease (including erythema, vascular pattern loss, alterations in mucosal appearance, and ulcers), and the percentage of a segment covered by any inflammatory CD feature. The CDEIS also requires differentiating between deep and superficial ulcers based on the different prognostic impacts of ulcers according to the ulcers’ depths, which has been confirmed in an independent study [9]. The SES-CD attempts to simplify the previously developed CDEIS, mainly by skipping the difference between deep and superficial ulcerations (which is meaningful but more likely to induce inter-observer variations) and considering their width as a more conservative approach. Moreover, the SES-CD measures and transforms an ulcerated/affected surface from a visual analog scale, required by the CDEIS, into a semiquantitative Likert scale with percentage ranges. Finally, ulcerated/non-ulcerated stenoses considered in the CDEIS are transformed in the SES-CD into a more functional concept of the presence or absence of narrowings based on the capacity to pass the scope through them in any ileocolonic segment.

The most intriguing and complicated step of both scores is the calculation of the percentage of an anatomic segment covered by a given lesion. This step requires an abstraction exercise in which the observer must imagine the tubular surface as an open rectangle, where the lesions of interest are summed to quantify the percentage of the total surface of the given ileocolonic segment they cover. The exercise is theoretical and probably influenced by the observer’s experience and sensation of the severity. This is likely an area of discrepancy between different observers (Figure 4) [32]. It is important to remember that the surface involved in any CD lesion should include the ulcerated surface, and, therefore, there cannot be a lower percentage for the affected surface than for the ulcerated surface.

The full CDEIS calculation requires all variables (the presence of deep or superficial ulcers and the surface affected by ulcers and any CD lesion) to be input for each segment. Then, the calculation considers how many segments are visible out of the maximum five segments. Thereafter, additional scores with weights are assigned to any ulcerated or non-ulcerated stenoses [27]. The SES-CD calculation is slightly more straightforward: each visualized segment requires the scoring of the lesions (ulcers, the ulcerated surface, the affected surface, and narrowings), all graded 0–3 points. The raw sum of all subscores in all explored segments leads to the final SES-CD score [28].

To maximize the performance of the SES-CD calculation for prognostic means, there were further refinements of the calculation algorithm for the SES-CD, leading to the so-called modified multiplier SES-CD (MM-SES-CD) [29]. The calculation can be performed using the McMaster University webpage at https://www.mcmasteribd.com/mm-ses-cd (accessed on 20 September 2023).

Using, recording, and documenting endoscopic disease activity, as well as using endoscopic scores, is crucial for measuring disease severity before a treatment decision and after any treatment. Scores help to compare such activity at different time points.

The advantages and limitations of the most commonly used endoscopic scoring systems for IBD endoscopic activity are summarized in Table 1.

In summarizing the evidence on using endoscopy for IBD severity assessment according to the ‘when’ and ‘how’ questions in this paper’s title, endoscopy is essential for IBD severity assessment.

**When?** Every time a major change in therapeutics is needed or when excluding a relevant endoscopic activity may impact therapeutic choices (but usually not scheduling endoscopy at precise time points).

**How?** Using as many reproduced and reproducible scores (the minimal standards are the MES and/or UCEIS for UC and the SES-CD and the Rutgeerts score for CD) as possible and documenting the most representative endoscopic features using videos/still images.

## 4. Using Endoscopy in Clinical Trials

The choice of which scoring system should be used depends on the clinical/trial setting and the precision of the estimate of endoscopic activity needed. Systematic endoscopic scoring systems are used to measure the impacts of therapeutic interventions in both clinical practice and clinical trials [33].

However, when using endoscopic score results in objectivate trial results, it should be noted that agreement, even between expert observers, is not perfect, and this may affect outcome evaluations [21,22,26,31,34,35].

Feagan et al. [34] showed that local readers tend to overjudge basal severity and underscore post-treatment disease activity. They showed that using a central reading system, a trial originally failing to demonstrate the efficacy of mesalamine for ulcerative colitis, would have led to positive results if patients over-judged as being more active by local readers had not been admitted to the trial. Even if the study population had been substantially smaller, the effect size of the active drug would have been maintained, and only the placebo response rate would have decreased, leading to a statistically significant result for the active drug. The FDA and EMA require that new IBD trials include endoscopic endpoints and a central reading system. Rutgeerts and colleagues [36] showed that, in Crohn’s disease, local SES-CD readers overscored basal and post-treatment endoscopic activity by 20–25% and by 10–15% using the CDEIS.

Central readers are advocated as experts, and studies show that they have a close correlation in scoring the same sets of videos [31,34] for both ulcerative colitis and Crohn’s disease. Moreover, their agreement performance is superior to that of inexperienced endoscopists [37]. However, training programs may help to increase agreement between less experienced endoscopists to close to that between central reviewers [21,22].

Discrepancies in scoring between central readers, or between central and local readers, may affect whether a patient is included in a trial or not or his/her amelioration qualifying or not for long-term maintenance or for reaching endoscopic endpoints. Efforts have been made to clarify the issues most affecting inter-reader agreement, and some points have been identified, specifically for CD scores [31]. The most important issues leading to relevant differences between central readers were in grading the severity of narrowing or stenoses, the interpretations of lesions at the anal verge (being scored to the rectum or not) or of lesions spanning across two adjacent segments or at the anastomotic site, and the interpretation of the depth of ulceration (for the CDEIS only).

Several reading models, including the adjudication of discordant scores [38], have been proposed to standardize central reading approaches and minimize the risks of under- or over-estimating endoscopic activity, and all may have an impact on trial designs and results [33]. Furthermore, central reading paradigms should be accurately considered when analyzing trial results.

Central reading companies should periodically document agreement between their readers and clearly state what kind of algorithm is used when discrepancies emerge between local and central readers, as they may have relevant impacts on study results. Moreover, local readers are the only people in contact with a patient and produce the original endoscopic clip; therefore, their inclusion in the process is essential.

In summarizing the evidence on using endoscopy in IBD clinical trials according to the ‘when’ and ‘how’ questions in this paper’s title, endoscopy is crucial for IBD clinical trial assessments.

**When?** Clinical trials usually include basal examinations (often used to qualify patients for the trials), early (usually 8–12 weeks after starting the treatment), and long-term (typically at 1 year) endoscopic activity evaluations, but different drugs and different study designs may require different timings of endoscopies.

**How?** A colonoscopy with a recording for central review evaluation is the standard. Local readers record the Mayo/UCEIS for ulcerative colitis and the SES-CD or Rutgeerts score for Crohn’s disease. Central readers confirm or dispute local readers’ scores, assigning scores to the same endoscopic videos.

## Figures and Tables

**Figure 1 diagnostics-13-03423-f001:**
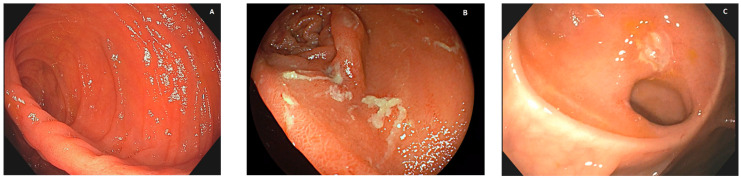
Different degrees of post-surgical recurrence. (**A**): No lesion is visible (no erythema, aphthae, or ulcers) at the site of ileocolonic anastomosis or upstream in the neoterminal ileum; therefore, the case can be assigned a Rutgeerts score of i0. (**B**): Several aphthae are visible, and there is extensive erythema of the interposed mucosa; therefore, the case is assigned a Rutgeerts score of i3. (**C**): When one or more ulcers are visible at the anastomotic inlet, the neoterminal ileum, or the ileal body, or if a stenosis is present, the case is assigned a Rutgeerts score of i4.

**Figure 2 diagnostics-13-03423-f002:**
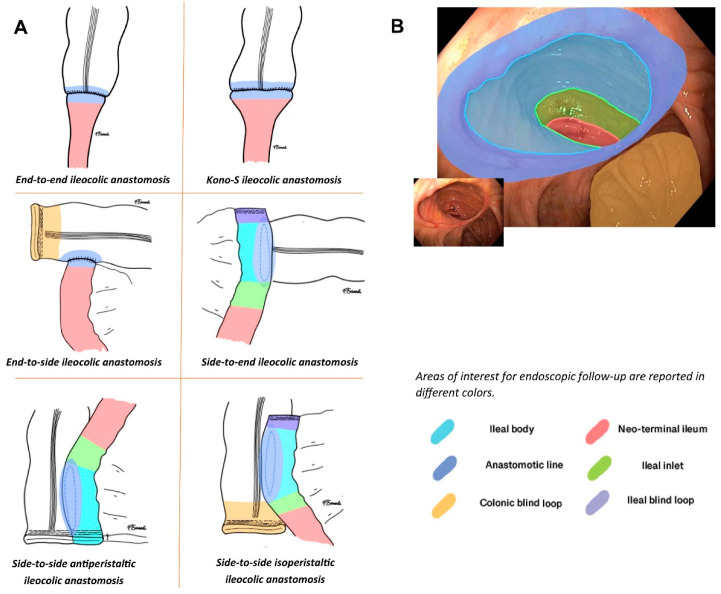
From Riviere et al.’s paper [20]. (**A**). Graphical representations of the different types of ileocolonic anastomoses. (**B**). Graphical overlay of color legend according to an endoscopic picture in order to differentiate different parts of the anastomotic anatomy. For different anastomotic shapes, the endoscopist should refer to different terminology. Areas of interest are the neoterminal ileum, the ileal inlet (the area approximately 1 cm from where there is the passage between the anastomotic body and the afferent ileal limb), the ileal anastomotic body, the ileal blind loop, the anastomotic line (where the ileal and colonic walls are in direct contact), and the colonic blind loop. Not all anastomotic shapes have all these elements, but considering the type of anastomosis, the report should refer elemental lesions to the correct areas.

**Figure 3 diagnostics-13-03423-f003:**
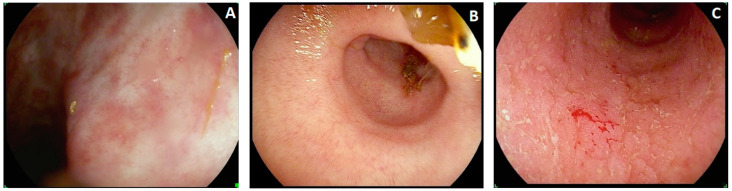
Examples and special features for determining Mayo endoscopic subscore (MES) for ulcerative colitis. (**A**): Presence of whitish mucosal scars, with uninflamed mucosa and vascular pattern sharply recognizable, is included in MES 0. (**B**): Mucosa still looks pinky, but light scattering in the top part of the picture is more fragmented, and the vascular pattern is not well recognized (but is still somewhat visible). This picture is typical of MES 1. (**C**): Erythema is diffuse, with tiny sub-millimetric mucosal breaks and yellowish fibrin deposits, which are the erosions, and some minor bleeding, likely resulting from contact with the scope due to friability. These features are typical of MES 2.

**Figure 4 diagnostics-13-03423-f004:**
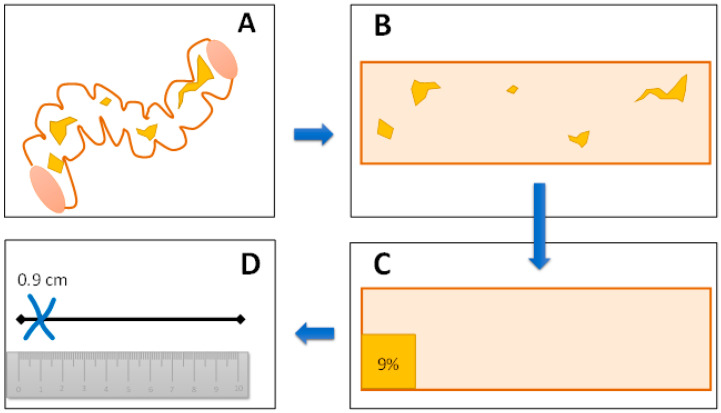
(**A**–**D**) The theoretical exercise an endoscopist should carry out to score the percentage of an ileocolonic segment affected by ulcers (the same applies to the affected surface, too) [32]. (**A**) Endoscopic exploration of an ileocolonic segment reveals ulcers of different widths at different points. (**B**) The endoscopist should imagine the segment as opened wide in a two-dimensional plane, such as after dissecting it across its longitudinal axis. (**C**) The observer should imagine the surface occupied by ulcers as a sum of areas compared with the total segment surface, leading to a percentage. (**D**) The observer can now draw the corresponding sign on a visual analog scale and measure it in centimeters (for CDEIS calculation) or compare it with the corresponding Likert scale (for SES-CD calculation).

**Table 1 diagnostics-13-03423-t001:** Comparison of strengths and limitations of most commonly used endoscopic scores regarding their precise activity reporting, responsiveness to therapeutic interventions, prognostic value, and usability in a central reading setting (adapted from [33]).

Score		Endoscopic Activity Reporting	Responsiveness to Treatments	Prognostic Value	Central Reading
**MES** [25]	**PROs**	Gross classification of inflammation Present standard for drug agencies (FDA and EMA) Defined by worst lesion	Development focused on responsiveness Extensively used over past 20 years in trials	Limited data on prognostic role in the literature	Reproduced algorithms for central reading (adjudication, paired reads, etc.) Categorical score leads to easier algorithms for adjudication Widely used over past 5 years
	**CONs**	Lacks precision for global burden of severity and extent of lesions Not properly validated Limited spectrum at lower and higher spectrum of activity Limited inter-observer agreement	Lack of ability to highlight segmental healing Lack of responsiveness due to four rigid grades	Not developed with prognostic intent	Limited inter-observer agreement May be inconsistent between readers due to little washing of the mucosa
**UCEIS** [26]	**PROs**	Extensive characterization of elemental endoscopic lesions focused on agreement	Wider diagnostic breadth than EMS		Already used in some trials
	**CONs**	Lacks precision for global burden of severity and extent of lesions Limited inter-observer agreement	Lack of ability to highlight segmental healing Limited use in clinical trials Development not focused on responsiveness	Not developed with prognostic intent	Agreement and adjudication are more complex for continuous scores compared with categorical scores Modest agreement on some lesions (e.g., bleeding)
**Rutgeerts** [11]	**PROs**	Clear-cut description of elemental lesions to be accounted for		Development focused on prognosis Many studies reproduced prognostic value of the score	Categorical score leads to easier algorithms for adjudication Algorithms for EMS easily exportable to Rutgeerts score
	**CONs**	Does not measure activity Does not include activity outside anastomotic site	No responsiveness evaluation	Developed for terminal–terminal anastomoses; never validated for side-to-side anastomoses Limited inter-observer agreement	Limited inter-observer agreement No data on paired reading
**CDEIS** [27]	**PROs**	Developed and validated to precisely report disease activity	Shown in few trials, even if not explicitly developed for responsiveness	Limited prognostic value extrapolated in retrospective studies	Used in trials
	**CONs**	Complexity Exact weight of each variable to be better clarified Relatively limited inter-observer agreement	Not developed with focus on responsiveness	Not intended for prognostic value	Limited inter-observer agreement Agreement and adjudication more complex for continuous scores compared with categorical scores
**SES-CD** [28]	**PROs**	Developed and validated to precisely report disease activity Possibility to easily exclude a given variable	Shown in several trials, even if not explicitly developed for responsiveness	Limited prognostic value extrapolated in studies	Widely used in trials; different algorithms available (fix or sliding scale for adjudication, paired reading, etc.)
	**CONs**	Relatively complex Exact weight of each variable needs to be better clarified Relatively limited inter-observer agreement	Not developed with focus on responsiveness	Not intended for prognostic value	Limited inter-observer agreement Agreement and adjudication more complex for continuous scores compared with categorical scores

MES: Mayo endoscopic subscore; UCEIS: ulcerative colitis endoscopic index of severity; Rutgeerts: Rutgeerts score for postoperative Crohn’s recurrence; CDEIS: Crohn’s disease endoscopic index of severity; SES-CD: simplified endoscopic score for Crohn’s disease; PROs: advantages; CONs: contraindications.

## Data Availability

Not applicable.

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
