# Peer review of "Endoscopy in IBD: When and How?"

_diagnostics, 2023, doi:10.3390/diagnostics13223423_

Round 1

Reviewer 1 Report

Comments and Suggestions for Authors

This review comprehensively elucidates the significance of endoscopic examinations in IBD  management. Not only does it discuss the role of endoscopy in diagnosis and disease severity assessment, but it also emphasizes its crucial importance as a predictive factor for prognostication. Moreover, it succinctly outlines the potential pitfalls encountered during these evaluations.

However, it also mentions the limitation, particularly in Crohn's Disease, where a standard endoscopic examination may not suffice for an adequate evaluation of small bowel lesions, suggesting that further discussion is warranted on the measures that should be adopted to address this issue.

Author Response

Thank you to the Reviewer for appreciating our efforts to clearly outline the reach and limitations of endoscopy for IBD

Reviewer 2 Report

Comments and Suggestions for Authors

This review article was produced by a well-known IBD expert Marco Daperno.   My concerns are as follows.

1. The current abstract included mostly general information regarding endoscopy for IBD. More concrete contents described in the main text would be better to be included in the abstract.

2. A table might be useful to introduce several scoring systems representing the endoscopic activity of IBD.

3. I wonder whether the styles of "When?" and "How?" in the end of each section were indicated by the editors, which seem to be a little odd.

Author Response

We thank the Reviewer for time and efforts spent at reviewing our chapter.

  1. the Abstract was rearranged according to the suggestions
  2. Endoscopic scores are reviewed extensively in the text, and summary table would allow mainly the list. However a Table 1 was added
  3. The outline with "When?" and "How?" repeated in all paragraph was a choice of the Authors